# A Tale of Wonders in Performance: The *Precious Scroll of Wang Hua* in the Storytelling Tradition of Changshu, Jiangsu, China

Rostislav Berezkin 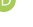

National Institute for Advanced Humanistic Studies, Fudan University, Shanghai 200433, China;
berezkine56@yandex.ru

**Abstract:** *Baojuan* (precious scrolls) are a type of prosimetric literature in the vernacular language that flourished in the lower Yangzi valley between the late nineteenth and early twentieth centuries. Most *baojuan* texts are devoted to religious themes, often involving wondrous figures and events which can be characterized as "supernatural". The *Precious Scroll of Wang Hua* (*Wang Hua baojuan* 王花寶卷) is a comparatively late text centered on the marvelous apparitions leading to the salvation of a lay person. It is a widespread text of the southern "scroll recitation" tradition as it survives in Changshu, Jiangsu, but to date, it has not received attention from scholars of Chinese popular literature and religion. Still, it is important for understanding the origins, development, and functions of precious scrolls and their contribution to the field of Chinese popular religion. The original text of the *Precious Scroll of Wang Hua* formed ca. end of the nineteenth century, but the present research mainly uses the manuscript version of a modern performer from the vicinity of Changshu (ca. 1995). This narrative combines two major topics of the wondrous manifestation of Bodhisattva Guanyin and the descent to Hell. Both topics can be traced back to the early "miracle tales". Here, they have been adapted to the local life and cultural setting. The figure of the skeptical and egoistic Wang Hua who initially rejected the injunctions of Guanyin is a type well known to the modern audiences of *baojuan*. Thus, the supernatural elements serve the purpose of reconfirming traditional beliefs and values in the contemporary society.

**Keywords:** miracle/wonder stories; precious scrolls; Chinese Buddhism; popular religion; prosimetric literature; ritualized storytelling; religious narratives

## 1. Introduction

*Baojuan* (translated as "precious scrolls") are a type of Chinese prosimetric literature in the vernacular language that appeared around the thirteenth or fourteenth century, but continued to develop in the later period, reaching the peak of popularity between the late nineteenth and early twentieth centuries, when these texts spread in the so-called Jiangnan region (the southern part of the Jiangsu province and the northern part of the Zhejiang province), where a variety of the Wu 吳 topolects are spoken[1]. While *baojuan* have been transmitted as written texts (both in manuscript and printed forms), they were mainly designed to be recited to broad lay audiences via the practice commonly known as "scroll recitation" (*xuan juan* 宣卷) in these areas. Presumably originating from Buddhist preaching, *baojuan* texts from Jiangnan include a number of topics and beliefs unrelated to Buddhism[2].

Yet, the contents of many precious scrolls, still popular in this region, preserve a connection with the Buddhist narrative subjects and beliefs, which remains unnoticed by many scholars. Here, I analyze the contents and cultural meaning of the *Precious Scroll of Wang Hua* 王花寶卷, a comparatively late text from the Jiangnan region which is centered on the traditional Buddhist topic of miraculous apparitions leading to the conversion and salvation of a lay person. It is a widespread text embedded in the "scroll recitation" traditions that still exist in southern Jiangsu. In Changshu 常熟 and Zhangjiagang 张家港,

they are also known as "telling scriptures" (*jiangjing* 講經) and are highly ritualized and preserve the archaic style of recitation (with the accompaniment using solely percussion instruments)[3]. The *Precious Scroll of Wang Hua* dates back to ca. late nineteenth century, and it was recently discovered in the possession of performers of *baojuan* in several areas around the city of Changshu (they are locally known as "masters of telling scriptures" [*jiangjing xiansheng* 講經先生], and they usually have a professional status). To date, the *Precious Scroll of Wang Hua* has not yet received attention from scholars of Chinese popular literature, although it is very important for understanding the adaptation and evolution of old Buddhist topics in contemporary folk storytelling and local religious culture. With the use of the *Precious Scroll of Wang Hua* as a case study, the present research is centered on multiple aspects of the functioning of the literary form of the precious scrolls in the modern period and their integration into the popular beliefs of the Chinese people.

The main purpose of this article is to demonstrate the special features of using narratives of the supernatural (wonder tales) in modern storytelling practices, and to uncover the significance of the *Precious Scroll of Wang Hua* to understand the function of the *baojuan* form and religious narratives in general. This text represents a traditional narrative with very old motifs that are comparatively rare in the modern practices of scroll recitation. It attests to the link between precious scrolls and "miracle tales" that have circulated in a Chinese Buddhist milieu since the medieval period, demonstrating the impact of traditional Buddhist narratives on the development of this type of literature even through a relatively late period. This link still has not received the necessary attention from the scholars of precious scrolls in China and in the West, though it is very important for the study of origins and sources of these texts.

## 2. The Text of the Precious Scroll of Wang Hua

The earliest known variant of the *Precious Scroll of Wang Hua,* as recorded in the Chinese catalogues of *baojuan,* is the manuscript copied by Yu Jun 于均, dated 1906 (Che 2000, p. 283). It is kept in the Library of Yangzhou University; judging by its special features, it comes from the Jiangnan region. One can suppose that the original text of this precious scroll was formed ca. end of the nineteenth century. According to the catalogue of *baojuan* compiled by Professor Che Xilun (now outdated and incomplete, as new collections of *baojuan* in China have been discovered since the time of its publication), this text is represented by ten manuscripts altogether, and six among them are dated to the period between 1906 and 1944, when scroll recitation flourished in the Jiangnan region (Che 2000, pp. 283–84). Some of these variants have the alternative names of *Precious Scroll of the Wise and Kind [Person]* (*Xian liang baojuan* 賢良寶卷) and *Precious Scroll, Admonishing Self-Cultivation* (*Quan xiu baojuan* 勸修寶卷) which underline the text's religious and didactic meanings.

I have not seen any evidence that the *Precious Scroll of Wang Hua* was ever printed by woodblock or lithographic printing methods in one of the Jiangnan cities, as often happened with other *baojuan* texts from this region between the late nineteenth and early twentieth centuries. Presumably, it existed only in the form of manuscripts that were used by the "masters of scroll recitation" (*xuan juan xiansheng* 宣卷先生) or "masters of telling scriptures" in this area. The *Precious Scroll of Wang Hua* is represented in several variants collected in the vicinity of Changshu, which suggests its comparative popularity in this area. Changshu and adjacent districts of Zhangjiagang city (before 1986 under the jurisdiction of Changshu County) preserve the traditional art of telling scriptures that was revived after the 1980s and still flourishes there.

The present research uses as its main source the manuscript of the *Precious Scroll of Wang Hua* that was copied by Yu Guanbao 虞關保 (b. 1931), a master of telling scriptures from the Xiaoshan 小山 village of the Gangkou 港口 Township in Zhangjiagang (originally belonging to Changshu County). It is dated to the *yihai* 乙亥 year, which apparently corresponds to 1995. Still, it is a traditional text of "telling scriptures" dating back to ca. early twentieth century. It was copied by Yu Guanbao from the texts of his teachers, local per-

formers of previous generations, which is a typical way of transmitting texts in this area. It is amazing that an old text has been used in this storytelling practice until recently.

The main contents of this text include several miraculous appearances by Guanyin, who tries to persuade the rich squire Wang Hua to engage in self-cultivation (meaning basic Buddhist practices and charity) to secure a better fate in the afterlife. According to the manuscript variant by Yu Guanbao, Wang Hua was an extremely rich elder from the Wujiang 吳江 County[4]. "His family [property] was abundant, rice was rotting in the storages, slaves and servant-girls in his house formed crowds; [he had] first and secondary wives, sons and daughters-in-law; his children and grandchildren filled the family hall. Wang Hua was already eighty-three years old, but he just enjoyed life and stayed in leisure every day; he did not think about self-cultivation and recitation of the Buddha's name." (Liang 2007, vol. 2, p. 817).

Attempting to convert Wang Hua, Guanyin descended to his house three times, each time taking a different form. The first time she appeared as a Buddhist monk (*heshang* 和尚), the second time as a Daoist priest, and the third time as another Buddhist cleric. However, Wang Hua did not accept these figures' instructions to leave the mundane world and engage in religious practices, and he always postponed the task of self-cultivation to a later time. Eventually, Yan-wang (King Yama), the lord of the underworld, sent his messengers to capture the soul of Wang Hua and deliver it to his court. In the face of death, Wang Hua asked his wives and descendants to go to the underworld in his place; however, all his family members refused. Wang Hua was dragged to the underworld, where he observed the sufferings of sinners. Then, he regretted his lack of belief, but it was already too late. However, Guanyin manifested herself in Hell, again taking the guise of a monk, and gave Wang Hua another chance to improve his behavior; he was returned back to life (*huan yang* 還陽). Then, he engaged in self-cultivation and persuaded his family members and friends to follow the path of the Buddha in order to avoid the afterlife torments in Hell.

There are also two variants of this text preserved in the collections of the masters of telling scriptures in the vicinity of Changshu. The first one, also titled the *Precious Scroll of Wang Hua,* was copied by Ma Xuefeng 馬雪峰 (b. 1972), a famous performer from the Zixia village 紫霞村 in the Baimao 白茆 District of Guli 古里 Township[5]. Another variant is the *Precious Scroll of Huang Hua* (*Huang Hua Baojuan* 黃花寶卷), a manuscript in the possession of Wu Linbao 鄔林寶 (reportedly born in the 1920s), another local master from the Daolin village 道林村 (now the Chalin village [查林村]) in the Liantang 練塘 Township of Changshu. Wu Linbao represents the tradition of telling scriptures that originated in the period before 1949; however, the exact date of his manuscript of the text is unknown. It was collected by Yu Dingjun 余鼎君 (b. 1942), a hereditary performer of telling scriptures from the Shanghu 尚湖 District in Changshu, in 2011[6]. Both variants were published in an edited form in a collection of precious scrolls from Changshu (Wu 2015, vol. 2, pp. 1489–503).

The first text is very close to the version represented in the manuscript of Yu Guanbao; it must come from the same source as the latter, representing a traditional text of scroll recitation in this area. It portrays Wang Hua as a carefree person who indulged in pleasures during the whole of his long life and realized his mistakes only when he ended up in the underworld after death (as is typical of precious scrolls and preceding religious narratives; see the next section). As for the *Precious Scroll of Huang Hua*, though the names of the protagonists and their stories in the two texts are similar[7], Huang Hua turns out to be a completely different person from Wang Hua. He is said to be a forty-year-old native of the Xinghua village 杏花村, Dantu 丹徒 County, in the Songjiang 松江 prefecture; his life is set during the Song dynasty (Wu 2015, vol. 2, p. 1497).

Huang Hua was a kind-hearted and hard-working man who could not take time for religious practices because he was always too busy with his household duties. This is why, though his soul was taken to Hell after his death, Guanyin pardoned him as a reward for his good deeds and transferred him to Heaven. Details of the miraculous descent of Guanyin in this text also differ: she first took the form of a beautiful young girl who persuaded Huang Hua to go to a mountain retreat, and the next time she appeared as a monk. The

first episode shows influence from other Guanyin stories, also narrated in several precious scrolls popular in the Changshu area (see the next section, below).

Thus, in the *Precious Scroll of Huang Hua* we see considerable variation in the contents of the text, representing the different perspectives of the later performers and editors of these texts. This must be related to a modern interpretation of traditional religiosity: for modern people, busy with a variety of things in everyday life, good deeds may be seen as more important than traditional religious practices such as the recitation of sutras or the invocation of the Buddha's name. The *Precious Scroll of Huang Hua* represents another version of the *Precious Scroll of Wang Hua*, used until recently in the storytelling practices of the Changshu area.

### 3. Origins and Analysis of the Wang Hua Story

The special features of the content and religious background of the *Precious Scroll of Wang Hua* are revealed in the first prosaic part of the narrative in the manuscript version of Yu Guanbao. While this says that Wang Hua came from the nearby Wujiang District, he is called "elder" or "householder" (*zhangzhe* 長者). This clearly shows Buddhist connotations, as in the Buddhist scriptures *zhangzhe* usually stands for lay believers and patrons of relatively high social status. In the Chinese translations of sutras, this word is often used to render the Sanskrit words *gṛhapati* and *śreṣṭhin*. Its meaning implies wealth, and in some contexts, it is connected with specific castes and professions; it may also mean the status of the head of a household (see Nattier 2003, pp. 22–25). Thus, the appellation connects this indigenous Chinese figure with the Buddhist texts and images of foreign origins; the story can be interpreted as a case of adaptation of Buddhist beliefs and concepts in China. In another manuscript version of the *Precious Scroll of Wang Hua* from Changshu (Ma Xuefeng's manuscript), the protagonist is just called a wealthy person.

If we approach the contents of this text from the point of view of the typology of Buddhist literature, we can characterize the plot of the *Precious Scroll of Wang Hua* as a combination of two major topics of such works: the wondrous appearance of Bodhisattva Guanyin, who descends to the mundane world in order to convert Wang Hua, and the journey to Hell, where Wang Hua himself observes the sufferings of sinners. Both are typical of narrative literature related to Buddhist beliefs in China since the early medieval period when Buddhism started to spread in this country.

The concept of multiple manifestations of Guanyin in the *Precious Scroll of Wang Hua* is related to the doctrine of "expedient means" (Ch. *fangbian* 方便, Skt. *upāya*), an important conception in Buddhist Mahayana philosophy[8]. Miraculous manifestations of bodhisattvas can be regarded as a particular case of "expedient means." Bodhisattva Guanyin, the Buddhist deity of mercy and compassion, is especially well known for appearing in various forms to convert people or to answer the pleas of believers. These beliefs are presented in the "Gates of Universal Salvation" (*Pu men pin* 普門品) chapter of the *Lotus Sutra of Wondrous Law* (*Miao fa Lian hua jing* 妙法蓮華經; Skt. *Saddharma-puṇḍarīka-sūtra*), an extremely popular Buddhist text in East Asia, as well as in other scriptures[9]. The "Gates of Universal Salvation" says that Guanyin, as a savior of living beings, can appear as any of thirty-three manifestations, according to the needs of believers[10]. This feature of Guanyin is also often portrayed in precious scrolls starting with the very beginning of the development of this genre[11].

Several precious scrolls from the late sixteenth to early seventeenth centuries center on the "wondrous responses" of Bodhisattva Guanyin (Guanshiyin)[12]. Apart from these, a number of similar narratives about Guanyin have seen a resurgence in popularity in the more recent period (nineteenth to early twentieth centuries). They are used in the practice of telling scriptures in Changshu even now. Among them are the popular texts such as the *Precious Scroll of Miaoying* (*Miaoying baojuan* 妙英寶卷)[13], the *Precious Scroll of the Family Hall* (*Jiatang baojuan* 家堂寶卷)[14], and the *Precious Scroll of the Watermelons* (*Xigua baojuan* 西瓜寶卷)[15]. All these are traditional texts, the earliest variants of which can be traced back to the early nineteenth century, while the origins of their subjects are even earlier.

These texts form the metatext of the *Precious Scroll of Wang Hua* and its modifications, as is typical of folklore and storytelling literature. The connections are especially conspicuous in the *Precious Scroll of Huang Hua,* in which Guanyin first appears in the guise of a young beautiful girl. This must be a reference to the stories of the the *Precious Scroll of the Family Hall* and the *Precious Scroll of the Watermelons*, centered around the motif of enlightening through seduction, being a special case of interpretation of "expedient means" in Chinese Buddhist narratives (see, e.g., Yü 2001, pp. 419–20, 426–38). This topic is not explicitly developed in the *Precious Scroll of Huang Hua*, though it can be deduced by comparing it with these other popular texts in the Changshu tradition. Apart from those, the *Precious Scroll of Huang Hua* mentions the story of Princess Miaoshan, which constitutes the subject of the *Precious Scroll of Xiangshan* (devoted to the origins of Guanyin), one of the most often performed texts in the Changshu tradition of "telling scriptures" today (Wu 2015, vol. 3, p. 1498)[16]. Guanyin plays an important role in the religious discourse of the *Precious Scroll of Wang Hua*, as her name is invoked in the concluding lines of this text (Wu 2015, vol. 2, p. 821).

Overall, the *Precious Scroll of Wang Hua* may be characterized as a prosimetric text based on a story of wonders, which is a common form of Buddhist narrative literature in China. These texts have been designated with various terms: *lingyan* 靈驗 (proofs of efficacy), *yingyan* 應驗 (proofs of response), or *ganying* 感應 (sympathetic response). Written down from the fourth century onwards, the records of wonders became very popular in the medieval period: they "testify for" the power of Buddhist deities, monks and nuns, and the efficacy of Buddhist devotional practices[17]. Though representing the popular level of Buddhist devotion, these stories were written down by literati, which serves as an additional testimony for the spread of Buddhism among the elite classes in the early period of religion's transmission in China.

While usually called "miracle tales" in the English-language research, these narratives of marvelous events represent a different concept from "miracles" in the Judeo-Christian religious context. The doctrinal background of this literature is characterized by the amalgamation of the imported Indo-Buddhist ideas with indigenous Chinese cosmological concepts. As Chiew Hui Ho put it, in these texts "the wondrous is underpinned by a correlative cosmology of an 'integrated, organic universe' and powered by *ganying* [感應], a concept that explains wonders as natural responses in a world of independent order, where things and people react to each other when certain conditions are fulfilled" (Ho 2017, pp. 1134–35). Still, many of these narratives are centered on the wondrous interventions of Buddhist deities that are comparable to "miracles" in the Western context, with the meaning of supernatural events[18]. The stories of wonders in the Buddhist context stand close to the "strange tales" *(zhiguai* 志怪), which became a widespread category of literature in the early medieval period[19]. The notion of "strange" here encompasses the things that can be characterized as "supernatural" from the modern reader's perspective (see Huntington 2001, pp. 119–22; also Campany 1996). The tale of Wang Hua is a noteworthy late example of a "miracle tale", representing supernatural intrusion into an ordinary life without a clearly expressed notion of response to a person's plea or devotion.

The stories of wonders related to major Buddhist scriptures and figures in general had a great impact on the development of precious scrolls, though this phenomenon is still understudied in Chinese and Western scholarship. There are examples of stories originating in the earlier Buddhist sources that developed into full-fledged narratives in the *baojuan* form. A famous example is the *Precious Scroll of Lady Huang* (*Huang shi nü baojuan* 黃氏女寶卷), known since the early sixteenth century. It apparently grew out of the story of wonders caused by a pious lady's recitation of the Diamond Sutra (*Jin gang jing* 金剛經)[20]. Similarly to the *Precious Scroll of Wang Hua*, it involves the motif of "journeying to the underworld", which attests to the punishments of sinners and rewards for the pious and kind-hearted.

The journeys to the underworld constitute a topic common in Chinese Buddhist narrative literature. This topic was very useful for preaching the doctrine, as the torments of

Hell were illustrations of the popularly interpreted concept of retribution (karma). Since the early period of the spread of Buddhism among the native populations of China, Buddhist monks often resorted to tales of retribution when they preached the doctrine. These sermons were described in the famous work *Biographies of Eminent Monks* (*Gao seng zhuan* 高僧傳; T. 2059) by the monk Huijiao (慧皎; 497–554 CE). These sermons were given in the middle of the night, when audiences were exhausted from the monotonous chanting of ritual texts, and they included exemplary stories, called *jātakas* (*bensheng* 本生) and *avadānas* (*yinyuan* 因緣) in this source, presumably related to the forms of foreign Buddhist literature. Huijiao also noted that the stories had a strong psychological effect on the audience:

> [W]hen the preacher speaks about impermanence[21], he makes the heart and body shiver with fear; if he speaks about hell, tears of anxiety gush forth in streams. If he points out earlier karma, it is as if one clearly sees one's deeds from the past; if he predicts future consequences, he manifests coming retribution. If he talks about the joys [of the Pure Land], his audience feels happy and elated; if he discourses on the sufferings [of hell], their eyes are filled with tears. 談無常則令心形戰慄，語地獄則使怖淚交零。徵昔因則如見往業，覈當果則已示來報。 談怡樂則情抱暢悅，敘哀感則灑淚含酸。 (T. 2059, vol. 50, p. 418a4-7)

Journeys to the underworld were widely featured in the "miracle tales" of the medieval period. Related stories occupied an important place in voluminous and famous collections such as the *Accounts of Mysterious Revelations* (*Ming xiang ji* 冥祥記) by Wang Yan 王琰 (c. 500 CE) and the *Records of Mysterious Retribution* (*Ming bao ji* 冥報記) by Tang Lin 唐臨 (600–?) (Liu 2011, pp. 590–96; Campany 2012, pp. 63–260; Gjertson 1989). A motif common to these stories is the sudden death and returning to life of a protagonist (either a monk or a lay person), who then leaves an eyewitness account of his journey to the underworld and the sufferings of sinners he observed there. These stories naturally possessed explicit didactic meaning[22]. For example, the *Accounts of Mysterious Revelations* by Wang Yan includes nineteen stories with this motif (Campany 2012, p. 45; see also, Teiser 1988a; Ermakov 1994, pp. 41–57; Schmid 2001, pp. 969–71). Thus, the episode of "returning back to life", present in the *Precious Scroll of Wang Hua*, has very old origins; in the premodern period it was widespread in China and indeed in the whole East Asian region (see, e.g., Nguyen 2022).

The topic of Hell in Buddhist literature is further developed in the indigenous *Scripture of the Ten Kings* (*Shi wang jing* 十王經), composed around the ninth century: the early manuscripts of this text were found in the sealed library of the Dunhuang cave monastery, discovered in 1900[23]. Significantly, it was also presented as the revelation of a Buddhist monk who once traveled to the underworld and then returned to life[24]. The motif of "underworld journeys" also often appeared in prosimetric literature in the vernacular language of the Tang (618–907) and Five Dynasties (907–960), some works of which also were discovered in Dunhuang. The most famous by far is the story of the monk Mulian (Skt. Maudgalyāyana) descending to Hell to look for the soul of his sinful mother, which is featured in a variety of texts from Dunhuang, the most prominent among them being the *Transformation Text on Mahāmaudgalyāyana's Rescue of His Mother from the Underworld* (*Damujianlian mingjian jiumu bianwen* 大目犍連冥間救母變文), represented in a number of manuscripts[25]. There are also the *Transformation Text of Hell* (*Diyu bianwen* 地獄變文; only partially survived) and the *Record of Emperor Taizong of Tang Visiting the Underworld* (*Tang Taizong ru ming ji* 唐太宗入冥記) (the manuscript copied ca. 906)[26].

The development of this literature was related to the proliferation of the visual images of Hell in this period, also appearing in the ritual setting (Teiser 1988a, pp. 457–59). A very popular text with the detailed description of the structure of Hell is the *Precious Manuscript of the Jade Registers* (*Yuli baochao* 玉歷寶鈔), usually included in the category of "morality books" (*shanshu* 善書), a type of didactic literature widespread in the late imperial period. It is claimed to be a revelation from the Song dynasty person, but it has been dated by Japanese scholars only to ca. sixteenth to seventeenth centuries. The earliest extant editions

date back to the beginning of the nineteenth century (see Livres de morale révélés par les dieux 2012, p. 102)[27].

The topic of the "underworld journeys" was very important in the precious scrolls from the earliest period of their development. The earliest extant example of narrative *baojuan* texts is the *Precious Scroll of Mulian* (the complete title being the *Precious Scroll of Mulian Rescuing His Mother [and Helping Her] to Escape from Hell and Be Born in Heaven* [*Mulian jiumu chuli diyu sheng tian baojuan* 目連救母出離地獄生天寶卷]), preserved as two incomplete manuscripts dated 1373 and 1440 (see Berezkin 2017, pp. 48–71). The journey to the underworld also occupies an important place in the aforementioned *Precious Scroll of Xiangshan*. Significantly, the *Precious Scroll of Xiangshan* also includes the episode of "returning back to life", as after Miaoshan's execution her soul descended to Hell, but Yan-wang did not find her guilty and let her free (see also, Dudbridge 2004, pp. 110–15).

The Hell themes became even more common in the precious scrolls of the seventeenth century, the period from which the texts dedicated specifically to the Ten Kings of Hell and other underworld deities survived (see Overmyer 1999, pp. 240–41). This type of precious scrolls developed in the later period, especially when their recitation became widespread in the Jiangnan region in the second half of the nineteenth century. The texts devoted to the Ten Kings of Hell are especially important in the tradition of telling scriptures in Changshu, as they constitute an integral part of funerary and memorial rites in this area (see Yu [2012] 2015, pp. 2587–91). Significantly, an episode with the underworld journey and "returning to life", which is used in several versions of the *Precious Scroll of the Ten Kings* (*Shi wang baojuan* 十王寶卷), still often recited in Changshu and the nearby city of Jingjiang 靖江 (on the northern bank of the Yangtze river)[28], can be traced back to the seventeenth century (Che 2009, p. 290; Overmyer 1999, pp. 240–41).

The funerary ritual in Changshu also includes a text of sectarian origins—the *Precious Scroll of Hell* (*Diyu baojuan* 地獄寶卷), dating back to the late sixteenth century, which details Luo zu's (羅祖, a patriarch of the Ming dynasty syncretic teaching) underworld journey (Che 2009, p. 394; Liang 2007, vol. 1, p. 218)[29]. The *Precious Scroll of Mulian* also is commonly used during funerary recitation in the Changshu telling scriptures (Berezkin 2017, pp. 155–63). In addition, an adaptation of the *Precious Manuscript of the Jade Registers*—*Precious Scroll of the Jade Registers* (*Yuli baojuan* 玉歷寶卷), which contains the extensive description of the ten courts of the underworld, has been integrated into the funerary telling scriptures there as well (Yu [2012] 2015, p. 2591). These texts dedicated to the Hell topics, locally known as the "underworld scrolls" (*ming juan* 冥卷), constitute an important part of the context and even the metatext for such narratives as that of Wang Hua in the tradition of Changshu telling scriptures.

The description of the netherworld in the *Precious Scroll of Wang Hua* is not very detailed, especially in comparison with the "underworld scrolls", which were specially dedicated to these topics. However, even in these succinct episodes of the journey to Hell we find references to specific concepts and rituals widely known in the Changshu area. These are, for example, the returning of the "loan of life" (*huan shou sheng* 還受生) and the "Blood Pond" (*Xue hu* 血湖). The first concerns the symbolic loan that is believed to have been made to a soul before the next rebirth, and the second concerns the afterlife punishment of women for their presumed ritual impurity. Both beliefs are reflected in the special precious scrolls recited for ritual purposes in Changshu. Thus, the text is well-integrated into the ritual discourse of telling scriptures.

One of the special features of the *Precious Scroll of Wang Hua*, in contrast with other similar *baojuan* still frequently recited in the Changshu area, is that it is narrated from the perspective of a common layman who travels to the underworld to witness the afterlife torments and punishment for disbelief. Unlike the religious figures who appear as the underworld travelers in other *baojuan* narratives used in the Changshu tradition, such as Mulian, Princess Miaoshan, and the Teacher of Non-Activism, Wang Hua's story appears as a real-life person's experience. Though the story is set in the late imperial period, Wang Hua's figure certainly stands very close to that of modern believers who are the main au-

dience for telling scriptures in Changshu. The agnostic, almost atheistic, attitude of Wang Hua at the beginning, and his preoccupation with his domestic affairs, clearly were very familiar to modern people. Still, his encounter with wondrous events and his temporary death experience that caused a spiritual transformation, could be very inspiring for modern audiences. It calls upon elderly people to abandon worldly affairs and to take up some kind of spiritual cultivation; the traditional approach in China (as well as in India). The topics of life and death, the illusory nature of world affairs, the danger of unrestrained desires, and the propagation of basic virtues, constituted an important part of the contents of telling scriptures in the Changshu area, which allowed it to attract more audiences and to survive and develop in the modern cultural environment.

At the same time, when we compare the *Precious Scroll of Wang Hua* with similar texts used in several traditions of scroll recitation in southern Jiangsu, we can see that the stories of punishments of unbelievers have been very common there. For example, the story of Sister Zhang from Meile 梅樂張姐 is popular in the telling scriptures of the Jingjiang area nearby. She declined to participate in, or help organize, the religious assemblies, and she mistreated Guanyin, who descended in order to convert her (as in the *Precious Scroll of Wang Hua*). This led to Zhang's punishment in the underworld: her soul was dragged to the Seventh Hall on the order of the Jade Emperor. After tortures in Hell, Zhang Jie was reborn as a worm. This story is included in the text of the *Precious Scroll of Ten Kings*, where it is put in the section of the Seventh Hall; therefore, it is popularly known as the "Case of the Seventh Hall" (七殿公文). While other narratives are often dropped in modern real-life practice in Jingjiang, this one is still often enacted by the storytellers, which demonstrates its cultural significance and popularity (Che 2009, pp. 339–41)[30].

A similar story of afterlife punishment appears in the *Precious Scroll of Wang Daniang* (*Wang Daniang baojuan* 王大娘寶卷) in the Changshu area, again including the manifestation of Guanyin[31]. This subject in precious scrolls also can be traced back to the late Qing period and is represented in a number of old manuscripts from the late nineteenth to early twentieth centuries, the alternative name of the text being the *Precious Scroll of Wang Daniang's Tour of the Underworld* (*Wang Daniang you diyu baojuan* 王大娘游地獄寶卷) (Che 2000, pp. 283, 342–43; see also, Pu 2005, vol. 16, p. 280).

Such stories as those of Wang Hua (Huang Hua), Wang Daniang, and Sister Zhang reveal an enhanced didactic meaning of telling scriptures in several areas of southern Jiangsu: Changshu, Zhangjiagang, and Jingjiang. These stories provide the religious context of the storytelling practices, using traditional texts of precious scrolls. The appearance of Guanyin in all these tales, acting not only as a merciful deity, but also punishing unbelievers and selfish and evil people, marks them as the narratives of Buddhist piety and wondrous advents, constituting a very important part of the contents of telling scriptures in these areas.

## 4. On the Use of the Subject of Wang Hua in the "Telling Scriptures" Practice

In the modern practice of telling scriptures in Changshu, the *Precious Scroll of Wang Hua* belongs to the category of "entertaining scrolls" (*xian juan* 閒卷, or *baixiang juan* 白相卷), according to the emic classification of performers[32]. This means that this text is intended to be recited at religious assemblies after major precious scrolls devoted to the popular nationwide and local deities. These, known as the "sacred scrolls" [*shen juan* 神卷]), vary with the specific occasion[33]. "Entertaining scrolls" were mainly in demand in the traditional period (before the 1980s), when the religious assemblies in this area required a very long time (usually a day and a night)[34]. According to the recollections of old performers, these texts were used to amuse the sleepy audience, who were tired from the night-long recitation of the sacred scrolls and ritual texts (*keyi* 科儀)[35] that accompanied various ritual actions[36]. This setting of recitation of narrative texts resembles that described in the early Buddhist sources, such as the *Biographies of Eminent Monks*, as quoted above.

In the 1980s, when telling scriptures started to revive, its basic arrangement changed. Now, assemblies held for the welfare of living people mainly take place in the daytime and

last for around twelve hours (6 a.m.–6 p.m.) with a lunch break, which has to do with the changes in social and cultural context of the recitation performances[37]. The need to entertain does not take precedence in the current situation, when other types of entertainment for the traditional audience are accessible. Entertaining scrolls usually are not used in the modern telling scriptures; only texts devoted to deities are recited, as attested by the performers from the Changshu area (Yu [2012] 2015, p. 2577). Still, the *Precious Scroll of Wang Hua* is not a purely entertaining text; it has other significant functions, namely didactic and indoctrinating. Therefore, occasionally it also appears in the program of telling scriptures in the modern period.

The didactic function of the *Precious Scroll of Wang Hua* is clearly pronounced in its introductory verses, which contain formulas and ideas, typical of the literary form of *baojuan*. The opening verse of the *Precious Scroll of Wang Hua* includes moral exhortation for people in the audience, urging them to be modest and to subdue their desires:

I admonish the wise and good people listen [to my words],

Please stop absurd craving and not changing your mind!

[Many] only aspire to become rich and honored to equal heaven,

Do not worry about life and death, and [those] poor and despised.

奉勸賢良大眾聽，切忌貪妄不回心。

至望接天長富貴，不愁生死與賤貧。 (Liang 2007, vol. 2, p. 817)

Furthermore, happiness in the present life is explained by the results of self-cultivation in the previous existence:

As you did not perfect [yourself] in the previous existence, now you are in suffering.

If you do not perfect [yourself] in the present existence, it will be also hard [for you in the future].

前世不修今受苦，今世不修卻也難。

This interpretation of karma (retribution) is widespread in Buddhist traditions, especially on the popular level, as it can be used to explain social inequality to the mass of believers. In many traditional societies where the Buddhist ideology was dominant, such as the Tibetan, the law of karma was considered to be what governed one's social position in subsequent rebirths (Ostrovskaia 2002, pp. 385–86). This belief is also directly expressed in several relatively widespread precious scrolls dating back to the late nineteenth century, such as the *Precious Scroll of Mulian* (see, e.g., Anonymous 1876, juan 1, pp. 35a–36b)[38].

At the same time, the introductory verse of the *Precious Scroll of Wang Hua* talks about the setting for recitation of this text, which corresponds to the ritualized atmosphere of modern telling scriptures:

The sun rises in the east and moves to the west,

I admonish you to recite the Buddha's name and worship Tathāgata[39].

It is not allowed to engage in casual talks and chat at all,

With the calm mind listen to scriptures and recite the name of Amitabha.

Please do not talk about family matters,

Now I will proclaim the precious scroll and discuss [it] in detail.

Listen to the recitation of the [*Precious*] *Scroll of Wang Hua*,

In all generations be faithful and kind, and be good people!

東方日出往西來，勸君念佛拜如來。

不可閑談並講話，靜心聽經念彌陀。

家常事情不要講，今宣寶卷細宣談。

聽宣一本王花卷，世世忠良做好人。

This emphasizes the main meaning of scroll recitation as a type of Buddhist ritual. It admonishes the audience to listen attentively and to not talk about casual things. It also refers to the special performance style of scroll recitation, when the refrain with the name of Buddha Amitabha is continuously chanted by the audience[40]. This style, while rooted in the old Buddhist practice of the Pure Land movements, has been characteristic of the precious scrolls since the very beginning of their development in the thirteenth to fourteenth centuries (Berezkin 2017, p. 54). It continues even now, assisting in the involvement of the audience, an important characteristic of folklore genres. Active members of the audience in Changshu form a kind of chorus, usually cooperating with the main performer, participating in all ritual assemblies that he (or she) presides.

The main function of the *Precious Scroll of Wang Hua* in the practice of telling scriptures is summarized in the concluding verses in the following way:

> The merit of proclaiming of the precious scroll is completed,
>
> The Buddhist gāthā of Wang Hua has reached the great conclusion,
>
> [All] good men and pious women in the [scripture] hall
>
> Will escape hell and leave the cycle of sufferings.
>
> As you are listening to scroll recitation, we admonish you to repent,
>
> And eagerly start self-cultivation and recitation of the Buddha's name!
>
> 宣揚寶卷功完滿，王花佛偈大圓滿。
>
> 在堂善男並信女，要免地獄出苦輪。
>
> 聽宣寶卷勸回心，願做修行念佛人。 (Liang 2007, vol. 2, p. 821)

This verse further underlines the religious significance of the text, which is designated as the "Buddhist gāthā"—a type of ritual verse used in the Buddhist sutras. Recitation of this type of text is ascribed with the religious merit (*gongde* 功德), which is an important notion in Chinese Buddhism. In the Buddhist context, this term usually stands for virtue, fortune, or goodness (Sansk. *guṇa*, *punya*), which is accumulated according to one's good actions. Since the early period, it was one of the central ethical concepts in Buddhism, as it was believed to be a way to improve one's karma and thus receive a better rebirth in the next life. On the popular level, it has been a key value for Buddhist believers in various national traditions (Harvey 2000, p. 18; Spiro 1982, p. 141). According to popular perceptions, copying or sponsoring a recitation of a Buddhist scripture can be "merit", which believers can use for their own profit or transfer to other persons (especially common for the purpose of the salvation of deceased relatives/ancestors)[41].

The concluding verse of the text also promises to improve the welfare of the participants of the assembly; it says that recitation brings fortune and averts disasters. These wishes are characteristic of *baojuan* texts, making the *Precious Scroll of Wang Hua* typical of this category of literature. The recitation of precious scrolls is equated to the recitation of sutras, thus its common name in Changshu is "telling scriptures", even though locals also are aware of the difference between these practices: unlike the services performed by the Buddhist monks, "telling scriptures" mainly encompasses storytelling. Still, as attested by interviews with performers and members from audiences of telling scriptures in Changshu, they understand it as a type of Buddhist service (*fo shi* 佛事), though the majority of stories in this type of performance (including the Wang Hua story) do not have connections with the Buddhist sutras, and many local deities of non-Buddhist origin are worshiped during these religious assemblies.

As we have already noted, Buddhist concepts occupy an important place in the religious context of the *Precious Scroll of Wang Hua*. The verses pronounced by Guanyin also contain warnings of the ephemeral and fleeting nature of human life, very typical of traditional Buddhist discourse:

> The moments [of life] are rapidly and easily passing,
>
> The sun and moon are like a running shuttle, they move day and night.

One single moment is worth one piece of gold.

With a piece of gold it is impossible to buy even one moment!

光陰迅速容易過，日月如梭曉夜行。

一寸光陰一寸金，寸金難買寸光陰。 (Liang 2007, vol. 2, p. 817)

The purpose of a believer in this life is thus the performance of good deeds and the engagement in religious practices, such as the recitation of scriptures and the chanting of the Buddha's name.

At the same time, the *Precious Scroll of Wang Hua* also had the pronounced characteristic of being quite entertaining, such that it could attract audiences. In the traditional period, telling scriptures served as a type of popular storytelling. The significant literary qualities of this text are revealed in its form. Narration in the *Precious Scroll of Wang Hua* takes place not only in the prosaic parts, but also in the poetic passages, the pairing of which constitutes one of the special features of the late narrative *baojuan*. Poetic passages do not repeat and amplify the contents of prosaic parts, as is common in *baojuan* of the early period (for example, the *Precious Scroll of Mulian Rescuing His Mother [and Helping Her] to Escape from Hell and Be Born in Heaven* and *Precious Scroll of Incense Mountain of Bodhisattva Guanshiyin of Great Compassion*), but rather introduce new events and facts. This concerns, for example, the first manifestation of Guanyin in Wang Hua's house:

Bodhisattva Guanyin has descended from clouds,

She turned into an old monk from the earthly world

And arrived to the gates of Wang Hua's house,

In order to persuade him to engage in self-cultivation.

Wang Hua was [already] eighty-three years old,

But he just craved for pleasures and did not perfect himself!

觀音菩薩下雲端，化作凡間一老僧。

來到王花墻門首，要勸王花去修行。

王花八十零三歲，只貪快樂不修行。 (Liang 2007, vol. 2, p. 817)

The same arrangement appears in the episode in which Wang Hua suddenly is summoned to the netherworld by Yan-wang (Liang 2007, vol. 2, p. 819). In general, the narrative of this text takes a natural flow, which can be called the developed form of prosimetrum, a common feature of Chinese vernacular storytelling literature. These formal characteristics can be traced back to the transformation texts of the Tang dynasty, discovered in Dunhuang, a good example being the *Transformation Text on Mahāmaudgalyāyana's Rescue of His Mother from the Underworld* (see also, Zeng 2003, pp. 121–22). Still, often repetition is present in the poetic passages of the *Precious Scroll of Wang Hua*, reflecting its main function, that is, recitation for broad audiences, in which women, then mainly illiterate, constituted the majority[42]. Repetition of key words and ideas assisted in better comprehension of the text offered through oral transmission.

As for the contents of the text, the *Precious Scroll of Wang Hua* is notable for the description of life in a wealthy Chinese rural estate during the early modern period. These details appear in the verse that Wang Hua chants to Guanyin as she visits him for the third time. They list major activities and traditional festivals that occur during all the twelve months of the year, for example, raising silkworms in the third lunar month, transplanting rice seedlings in the fifth lunar month, gathering fruits in the eighth lunar month, gathering late rice, and grinding grain and storing it in the granary in the ninth lunar month. The most detailed description can be found in the *Precious Scroll of Huang Hua* (Wu Baolin's manuscript), where each month is portrayed in a corresponding couplet. Several details in these passages in all three versions of the tale represent local characteristics of the Changshu (and broader Suzhou) area. These include, for example, the process of making and the first use of a boat, as water transportation was then the most common means of travel in

this region; certain kinds of fruits and crops specific to this area, were also celebrated. The mention of sericulture also is characteristic, as it is widespread there.

Thus, in the *Precious Scroll of Wang Hua*, the wondrous events of the manifestations of Guanyin are related to the everyday life experience of peasants in the Suzhou area in the premodern period. Descriptions such as are found in this text are comparatively rare in the area's precious scrolls. They contribute to the cultural and historical value of this text. While it is generally accepted by Chinese scholars that precious scrolls reflect the cultural memory of the local people in a given area (Y. Li 2016), many texts are not very detailed in these matters[43]. The *Precious Scroll of Wang Hua* certainly stands out in this respect.

The verses on Wang Hua's activities throughout a year are arranged according to the "twelve-months" scheme, which is related to the popular tune used in folk songs as well as telling scriptures in this area (Liang 2007, vol. 2, p. 5). It has very old origins in the popular musical culture, as it can be traced to the sources of the late imperial period, and it is used in many dramatic and musical storytelling genres across China (Lin 2011, pp. 131–41). This further connects the text of the *Precious Scroll of Wang Hua* with local folk culture.

As for the other entertaining features of this text, the dialogues between Wang Hua and his family members in the Yu Guanbao's version of the *Precious Scroll of Wang Hua* have an almost comical effect. Needless to say, precious scrolls in the tradition of Changshu's "telling scriptures" have been recited using the local language, which is reflected in a number of dialectal words and expressions in the written text, for example, *wangtang* 王堂 for "county magistrate", *xiaosha* 笑煞 for "die laughing", *kaisui* 開歲 for "beginning of the year", *a youchu* 阿有處 for "what can we do", etc. These words were used not only in the original manuscript but also were preserved in the edited text of the precious scroll as it appeared in the "Collection of *Baojuan* from Heyang", published in Zhangjiagang. Not all of these words can be found in the dialect of Suzhou city, so presumably they represent the local dialect of Changshu. Dialectal words are especially common in the dialogues, for example: "Wang Hua said: 'Monk, let me tell you, and you will listen [to me]!'" (王花道：和尚吾來說拔儂聽聽吧). These linguistic features attest to the local characteristics of the text. This means that wherever the original text of the *Precious Scroll of Wang Hua* was composed, it was completely integrated into the local culture of the Changshu area by the modern period.

## 5. Conclusions

The *Precious Scroll of Wang Hua* is a testimony to the preservation and development of traditional Buddhist topics in *baojuan* literature from the Jiangnan region in the late period (late nineteenth to early twentieth centuries). While in Western scholarship precious scrolls have been primarily associated with the new religious teachings, proclaimed heterodox by the imperial state (which is true for the period of the sixteenth to seventeenth centuries), one needs to note that many late examples, such as this text, generally propagated mainstream ideas and values and preserved close ties with the earlier Buddhist narratives. The *Precious Scroll of Wang Hua* proves that the Buddhist wonders (also termed miracles) were an important theme, continuously present in the texts of this form, from the early to late period of its development; this still has not been adequately presented in the existing Chinese and Western research.

The *Precious Scroll of Wang Hua* combines two traditional topics, namely, the wondrous appearance of Bodhisattva Guanyin descending to the mundane world to convert lay people, and the journey to Hell, where the protagonist observes the sufferings of sinners. These can be traced back to the early Buddhist narrative literature in China, including the so-called "wonder tales", and also were used continuously in the prosimetric storytelling texts of the eighth to seventeenth centuries. These narrative motifs are related to the core Buddhist concepts of retribution (karma), "expedient means" and "numinous response." They have been common in religious literature since the early period of Buddhism's spread in China.

The "wonder tales" usually are interpreted as representing the popular form of Buddhist beliefs in China. The *Precious Scroll of Wang Hua* is also clearly addressed to the lay believers of the grassroots level, preaching basic Buddhist ideas and values. The menace of afterlife torments in Hell serves the didactic function, typical of precious scrolls overall. Furthermore, the *Precious Scroll of Wang Hua* advocates the necessity for religious practice embodied in telling scriptures. It can be treated as an example of the "lay" forms of Buddhist devotion, which are also well integrated into the field of "popular religion" of the Chinese. The present case study of the *Precious Scroll of Wang Hua* demonstrates an important role of the Buddhist elements in the modern forms of Chinese popular religion, which is often downplayed in the relevant Western and Chinese research.

While the original variant of the *Precious Scroll of Wang Hua* can be traced back to the end of the nineteenth century, at the latest, it has been used in the "telling scriptures" traditions of Changshu and Zhangjiagang until recently, which demonstrates the persistence of traditional narratives in the folk storytelling practice of the given areas. In addition, there were also modifications made to the modern variants of this story, represented by the manuscripts of local professional storytellers—"masters of telling scriptures". With the development of the key storyline in the new recensions of this precious scroll, the legend of Wang Hua became fully integrated into the local culture of the Changshu area, as shown by the fact that it includes numerous dialectal expressions as well as valuable information on the everyday activities and customs of local people in the traditional period. The text also combines didacticism and the preaching of basic Buddhist values with entertaining features that once played an important role in the functioning of this type of storytelling. Thus, the Buddhist wonders and admonitions came to appear broadly in the area's secular life.

**Funding:** This research was assisted by grants from the State Social Sciences Foundation of China: "Survey and multi-disciplinary research on the sources of performing arts related to the folk beliefs in the Taihu-lake area" (太湖地區民間信仰類文藝資源的調查和跨學科研究, 17ZDA167) and the "Survey and research on Chinese precious scrolls preserved abroad" (海外藏中國寶卷整理與研究, 17ZDA266).

**Institutional Review Board Statement:** Not applicable.

**Informed Consent Statement:** Not applicable.

**Data Availability Statement:** Not applicable.

**Acknowledgments:** The author expresses his gratitude to Yu Dingjun for providing materials; to two anonymous reviewers for their comments; and to Paula Roberts for her editorial work.

**Conflicts of Interest:** The author declares no conflict of interest.

### Abbreviations

T: *Taishō shinshū Daizōkyō* 大正新修大藏經, edited by Takakusu Junjirō 高楠順次郎 and Watanabe Kaigyoku 渡邊海旭. Tokyo: Taishō issaikyō kankōkai, 1924–1932.

## Notes

[1]　For an introduction to *baojuan* texts, see, e.g., Sawada (1975); Overmyer (1999); S. Li (2007); Che (2009).

[2]　The religious background of precious scrolls in the modern period can more accurately be characterized as the "popular religion" of the Chinese, which combines elements from various religious traditions (notably Buddhist and Daoist movements), while also possessing distinct local characteristics. On the definition of this term in application to Chinese religious history; see, e.g., Teiser (1988b, pp. 15–18).

[3]　On "telling scriptures", see, e.g., Yu (1997, [2012] 2015); Che (2009); Qiu (2010); Berezkin (2013). Here, I do not provide information on its history or ethnographic descriptions of its modern setting.

[4]　Wujiang County is located in the vicinity of Suzhou City. It is also a place where *baojuan* recitation is still widespread; see Che (2009, p. 322). In another variant of this text from Changshu, Wang Hua lived in the Wuxian 吳縣 County of the Hangzhou 杭州 Prefecture, but this is obviously incorrect: Wuxian is a historical name of Suzhou.

5    He started to study telling scriptures at the age of seventeen, with older local performers, and was known for his good vocal qualities as well as his considerable collection of *baojuan* texts; see Wu (2015, vol. 3, p. 2551).

6    On Yu Dingjun and his activities centered on the revival and modification of traditional scriptures, see Berezkin (2013).

7    The similarity of pronunciation of these two surnames (very widespread surnames in China) can lead to their interchange in literary works; note, for example, the story of lady Huang/Wang, a popular subject of the *baojuan* texts.

8    According to the Buddhist scriptures, the Buddha tried to attract the attention of listeners and lead them to enlightenment with the words and images with which they were familiar; see, e.g., Schroeder (2001, pp. 9–37).

9    This scripture has been translated into Chinese several times, the most famous and commonly used translation being that of Kumarajiva, made ca. 406 (T 262, vol. 9. 1c162b). The earliest extant translation of the *Lotus Sutra* by Dharmarakṣa (Zhu Fahu 竺法護) dates to ca. 286 C.E. (T 263, vol. 9. 63a133b).

10   Another popular Mahayana scripture, the Śūraṅgama Sūtra (*Shou lengyan jing* 首楞嚴經, T. 945) lists thirty-two manifestations of Bodhisattva; see Yü (2001, pp. 45–48).

11   Guanyin also manifests in several forms in the *Precious Scroll of Xiangshan* (*Xiangshan baojuan* 香山寶卷), another very famous early text, the original version of which dates back to the fourteenth or fifteenth century. These include Princess Miaoshan, a monk, and a hermit of the Xiangshan mountain. This is predictable in view of the close connections between this text and the *Lotus Sutra*; see Berezkin and Riftin (2013).

12   These are the *Precious Scroll of Bodhisattva Guanyin Sending a Baby to be Reincarnated, with the Complete Explication* (*Xiaoshi Baiyi Guanyin pusa song yinger xia sheng baojuan* 銷釋白衣觀音菩薩送嬰兒下生寶卷), presumably composed in 1582, and the *Precious Scroll of Bodhisattva Guanshiyin, Rescuing from Sufferings and Hardships* (*Jiu ku jiu nan linggan Guanshiyin baojuan* 救苦救難靈感觀世音 寶卷), dating back to ca. early seventeenth century, which is based on the canonical text of the "Gates of the Universal Salvation" chapter, elaborating on the wondrous responses of Guanyin to her believers; on them, see Yü (2001, pp. 465–74).

13   The alternative title is the *Precious Scroll of Guanyin in White Clothes* (*Baiyi Guanyin baojuan* 白衣觀音寶卷).

14   The alternative title is the *Precious Scroll of Guanyin with the Fish Basket* (*Yulan Guanyin baojuan* 魚籃觀音寶卷).

15   For the various recensions collected in the Changshu area, see Wu (2015, vol. 1, pp. 173–89); Liang (2007, vol. 1, pp. 251–59) and Zhonggong Zhangjiagang shiwei xuanchuanbu (2011, vol. 2, pp. 994–1004); Wu (2015, vol. 1, pp. 241–48); Liang (2007, vol. 1, pp. 171–77) and Zhonggong Zhangjiagang shiwei xuanchuanbu (2011, vol. 2, 1170–75); Wu (2015, vol. 2, pp. 1217–27); Liang (2007, vol. 1, pp. 243–50) and Zhonggong Zhangjiagang shiwei xuanchuanbu (2011, vol. 1, 522–28); see also Yü (2001, pp. 259–60, 419–32, 434–37).

16   This is a very early narrative *baojuan* (ca. fourteenth to fifteenth century), though it survived only in the form of editions dating back to the eighteenth century. For the earliest version of this text, reprinted in Hanoi in 1772, with the complete title of the *Precious Scroll of Incense Mountain of Bodhisattva Guanshiyin of Great Compassion* (Ch. *Dabei Guanshiyin pusa Xiangshan baojuan* 大悲觀世音菩薩香山寶卷), see Berezkin and Riftin (2013); for the translation of the later recension, see Idema (2008, pp. 45–159); on the development of the Miaoshan story in Chinese literature, see Dudbridge (2004, pp. 5–78); Yü (2001, pp. 338–52).

17   For the history of Buddhist tales of wonders/miracles in China, see Gjertson (1989), Liu (2011, pp. 579–600), Campany (2012, p. 562).

18   See also, Zhiru (2007, p. 168). There is also the Daoist tradition of tales of wonders, also going back to the very early period, which represents another aspect of Chinese narratives on the "supernatural"; I do not deal with it here.

19   Often, stories of Buddhist wonders/miracles are regarded as a subtype of the *zhiguai* literature in modern research: Huntington (2001, pp. 121–22); Liu (2011, pp. 579–600); Campany (2012, pp. 17–29).

20   See Sawada (1975, p. 158); early recensions of this text apparently did not survive; see also, Che (2009, pp. 124–26); Grant and Idema (2011, pp. 11–17). For the trans. of the earliest available version of this tale in the *baojuan* form, as partially included in the late sixteenth-century novel *Jin Ping Mei* 金瓶梅, see Idema (2021, pp. 195–205).

21   "Impermanence" in the Buddhist context also refers to death.

22   The notion of retribution had a great impact on the development of Chinese traditional literature, remaining "one of the central subjects and organizing principle of both classical and vernacular fiction until late in the Qing dynasty"; Huntington (2001, p. 112).

23   The complete title of this text is the *Scripture Spoken by the Buddha on the Prophesy to the Four Orders of King Yama Concerning the Seven Feasts to Be Practiced Preparatory to Rebirth in the Pure Land* (*Yanluo-wang shouji si zhong ni xiu sheng qi wang sheng Jingtu jing* 閻羅王授記四眾逆修生七往生淨土經). Another variant of this apocryphal (indigenous) Chinese scripture was preserved in Japanese collections; see Teiser (1994).

24   As the rituals invoked in the *Scripture of the Ten Kings* have roots in Daoist practice, it must be characterized as the text of "popular religion", encompassing beliefs shared by several religious traditions.

25   For the edited version of this and other similar texts on Mulian from Dunhuang, see Huang and Zhang (1997, pp. 1024–76); for the English trans., see Mair (1983, pp. 123–66); see also, Mair (1989, pp. 14–15, 17–18, 123–27).

26    See Huang and Zhang (1997), pp. 319–22; see also, "The Record of a Returned Soul" (*Huan hun ji* 還魂記), discovered among the Dunhuang manuscripts; Teiser (1988a, pp. 448–49).

27    For the English trans., see Clarke (1894); see also, Duyvendak (1952).

28    Jingjiang also has a tradition of telling scriptures, for which see, Che (2009, pp. 279–333); Lu and Che (2008); Huang (2013).

29    Patriach Luo, or Ancestral Teacher of Non-Activism (Wuwei zushi 無爲祖師), in this text presumably refers to Luo Qing 羅清 (ca. 1442–1527), the founder of the Teaching of Non-Activism and author of several precious scrolls; though the *Precious Scroll of Hell* represents another religious tradition.

30    Another similar story is recorded in the same passage of the *Precious Scroll of the Ten Kings* as included in the collection of Jingjiang baojuan: punishment of Meile zhangzhe 梅樂長者 (You et al. 2007, vol. 1, pp. 444–46: which storyteller's version is not indicated).

31    For the variant based on the undated modern manuscript of Yang Meilan 楊美蘭 from the Dengying 登瀛 village of the Jinfeng 錦丰 Township, in Zhangjiagang, see Liang (2007, vol. 2, pp. 1025–32); see also, Zhonggong Zhangjiagang shiwei xuanchuanbu (2011, vol. 2, pp. 640–47).

32    白相 (Suzhou dialect: bəʔ³siã²¹; "play, leisure") is a common word in the Wu group of dialects.

33    I do not provide here ethnographic descriptions of "telling scriptures" sessions; the interested reader can turn to earlier published scholarship.

34    I define the period from ca. 1900 to 1980 as the traditional one; however, since the 1950s, telling scriptures was generally forbidden, and the practice went underground, especially during the Cultural Revolution. The modern period of the history of this art starts with its revival in the 1980s.

35    Sometimes also translated as "litanies" in the English-language scholarship.

36    The similarity of the arrangement is described by Li Shiyu, who conducted fieldwork research on the Wu-area scroll recitation in the 1940s; see S. Li (2007, p. 22).

37    The arrangement for funerary recitation is different, as is its subject matter; see Yu ([2012] 2015, pp. 2587–91).

38    For the complete English translation of this text (the 1885 edition) by Wilt L. Idema, see Grant and Idema (2011), pp. 35–145.

39    *Tathāgata* (Ch. Rulai), meaning "one who has thus come/gone", is an appellation of the Buddha.

40    The text of the *Precious Scroll of Huang Hua* (Wu Linbao's manuscript) indicates that the name of Bodhisattva Guanshiyin also must be invoked before the singing of every poetic passage; see Wu (2015, vol. 2, p. 1497). This feature further emphasizes devotional meaning of this text and also can be compared with the similar invocations of her name in the poems of the *Precious Scroll of Xiangshan*, one of the most often performed texts in the Changshu area; Anonymous (1886, p. 2a).

41    The notion of merit is widely propagated, e.g., in the *Lotus Sutra*, which is closely related to the cult of Guanyin, see, e.g., T. 262 (vol. 9, p. 19B–23).

42    Fieldwork observations demonstrate that the audience for telling scriptures is mainly composed of female believers at present as well.

43    Notable exceptions are the aforementioned precious scrolls from the Jingjiang area, which have been transmitted primarily in the oral mode in the recent period. Many of these voluminous texts contain details of local life and customs; see, e.g., Huang (2013).

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
