# Peer review of "A Tale of Wonders in Performance: The Precious Scroll of Wang Hua in the Storytelling Tradition of Changshu, Jiangsu, China"

_religions, doi:10.3390/rel14050670_

Round 1
Reviewer 1 Report
This is a fine article on a preeciosu scroll that so far has not attracted scholarly attention. The author clearly links the main topics of the text to basic notions of Buddhism and stresses the relevance of its contents to a modern audience. The article is clearly written but would benefit from copyediting by a native speaker of English. The author clearly is highly competent in this area of research and well acquainted with the relevant scholarship.
Two small comments: line 360. How can a single person be "reborn as a swarm"?
If Wang Hua is described as a very rich man, can he be characterized as a "peasant"?
Author Response
1. it was a typo
2. There were also quite rich peasants (landowners) in traditional China. I have added an explanation.
Reviewer 2 Report
See attached PDF.

Author Response
I was glad to read general approval of my research.
Here is my response:
- Berezkin is cited in this article.
- I have re-written the introduction and conclusion to demonstrate significance of this study and its broader implications.
- The article was proofread and edited by the professional editor of English.
- Southern tradition of recitation refers to scroll recitation in Jiangnan ( southern Jiangsu specifically), known for its ritual context
- Rephrased
- Changed to “self-cultivation”
- Clarified, it means a Buddhist monk
- I have changed to Yan-wang everywhere to be consistent
- I have noted this is a recurring motif in baojuan and earlier Buddhist narratives
- I have noted these are recurring themes in the texts about Guanyin in late imperial China, including baojuan
- Anonymized refers to my own articles, not anonymous texts
- This was a wrong word, I have corrected it.
- I have added explanation of my periodization of “traditional” and “modern” telling scriptures.
- I have added my interpretation of the inclusion of everyday activities and major local festivals in the text of this baojuan.
- I have re-phrased this sentence, and re-wrote the whole conclusion.
Reviewer 3 Report
The subject matter discussed in this manuscript is interesting. However, there are several issues that need to be refined as follows.
1. Many expressions in the manuscript are challenging to understand, especially some technical terms without corresponding Chinese annotations, which will cause trouble for English readers. Therefore, the manuscript should be professionally revised in terms of language.
2. The manuscript analyzes the contents of the Precious Scroll of Wang Hua and its relation to Buddhism and daily life. However, it lacks an analysis of the particularities and localities of the Precious Scroll of Wang Hua in the context of the whole baojuan 宝卷. In other words, where the uniqueness of the Precious Scroll of Wang Hua requires the author to continue his in-depth analysis.
3. The manuscript concentrates on the content of the Precious Scroll of Wang Hua and the environment in which it was used, which is undoubtedly valuable. However, there is a greater need to highlight the contribution of this case to the study of the baojuan 宝卷 as a whole and the study of Chinese folk beliefs.
Author Response
- Editing has been completed.
- I have added explanations of the unique characteristics of the Precious Scroll of Wang Hua.
- I have emphasized the contribution of this case study to the study of the baojuan texts and Chinese folk beliefs in general: mainly connections btw. Buddhist beliefs and narratives in the baojuan texts of the later period, continuation of the old Buddhist motifs in the late narratives as the Precious Scroll of Wang Hua and their adaptation to the life and culture of commoners in the modern period.
Round 2
Reviewer 3 Report
The manuscript has been meticulously revised, but could be further strengthened in terms of addressing the contribution of this study to the study of the Baojuan and Chinese folk religion.
Author Response
I have revised the manuscript (especially concluding and introductory parts) in order to emphasize its contribution to the study of the Baojuan and Chinese folk religion. As the topic of this special issue is the narratives of wonders in China, I have mainly addressed the issue of how the traditional themes of Buddhist wonders were integrated into the late narrative that has been used in the traditional-style ritualized storytelling practice of the southern Jiangsu areas. This involves the issue of the use of miracle stories in the formation and development of baojuan narratives that has not been fully addressed in the existing studies of baojuan in Chinese and other languages. Besides, this case study proves the multi-functionality of the baojuan narratives in the traditional cultural context of the Changshu recitation. This is the main contribution to the study of baojuan.
This article also demonstrates how the traditional narrative originating ca. end of the nineteenth century is used in the modern context of storytelling practices, which attests to the persistence of the traditional religious subjects in this type of art. The story of Buddhist wonders was integrated into the local culture and life circumstances of the local people. Besides, there were modified variants of this story that brought it closer to the level of modern believers. This demonstrates an important role of the Buddhist elements in the modern forms of Chinese popular religion, which is often downplayed in the relevant Western and Chinese research. This makes the contribution to the study of Chinese popular religion (or folk beliefs) in the modern period.